# Efficacy of Ipilimumab and Nivolumab in Patients with Melanoma and Brain Metastases—A Danish Real-World Cohort

**DOI:** 10.3390/cancers16142559

**Published:** 2024-07-17

**Authors:** Karoline Dreyer Kattenhøj, Christine Louise Møberg, Louise Mahncke Guldbrandt, Rasmus Blechingberg Friis, Christophe Kamungu Mapendano, Søren Kjær Petersen, Christina Halgaard Bruvik Ruhlmann, Inge Marie Svane, Marco Donia, Eva Ellebaek, Henrik Schmidt

**Affiliations:** 1Faculty of Health, Aarhus University, 8000 Aarhus, Denmark; 2Department of Oncology, Aarhus University Hospital, 8200 Aarhus, Denmark; 3Department of Oncology, Aalborg University Hospital, 9220 Aalborg, Denmark; 4Department of Oncology, Odense University Hospital, 5000 Odense, Denmark; 5Department of Clinical Research, University of Southern Denmark, 5230 Odense, Denmark; 6Center for Cancer Immune Therapy (CCIT-DK), Department of Oncology, Copenhagen University Hospital, Herlev and Gentofte, 2730 Herlev, Denmark

**Keywords:** ipilimumab, nivolumab, melanoma, brain metastases, DAMMED, immunotherapy

## Abstract

**Simple Summary:**

This retrospective database review of Danish real-world patients with asymptomatic melanoma brain metastases (MBM) treated with first-line ipilimumab and nivolumab demonstrates a similar complete response (CR), Progression-Free Survival (PFS), and overall survival (OS) rates when compared to previously published phase II trials.

**Abstract:**

Combination immunotherapy using ipilimumab/nivolumab is the golden standard treatment for patients with melanoma and asymptomatic brain metastases (MBM). However, it remains uncertain if real-world patients have the same treatment effects compared to patients enrolled in clinical trials. The aim of this study was to compare clinical benefits between real-world patients and patients enrolled in clinical trials when administering ipilimumab/nivolumab in treatment-naive patients with asymptomatic MBM. Using data from the Danish Metastatic Melanoma Database (DAMMED), 79 patients with clinical parameters similar to the inclusion criteria from two phase II trials, the ABC and the CheckMate-204 trials, were included in the analyses. Thirteen patients (16.5%) achieved complete response (CR) and an overall response rate (ORR) of 46.9%. We found an overall 6-month Progression-Free Survival (PFS) rate of 53.5% and a median PFS of 6.5 months. Median overall survival (mOS) was not reached during the 5-year follow-up. These results were comparable to the phase II trials. In conclusion, clinical benefits from phase II studies were comparable to Danish real-world data regarding OS, PFS, and CR. Confirming that combination immunotherapy can be recommended as first-line treatment for patients with asymptomatic, treatment-naive melanoma brain metastases.

## 1. Introduction

Brain metastases in patients with melanoma are common, with approximately 50% of patients with melanoma developing brain metastases [1]. This was previously a difficult complication to treat due to the lack of effective treatment options. Historically, patients diagnosed with melanoma and brain metastases (MBM) have had a poor prognosis and have often been excluded from clinical trials, such as in the original Checkmate-067 trial [2]. This trial gave precedence for the use of combination immunotherapy, ipilimumab/nivolumab (ipi/nivo), on patients with melanoma without brain metastasis, as the study showed a greater overall survival for patients receiving ipi/nivo compared to patients receiving ipilimumab monotherapy. 

However, the results of the ABC study [3] and the CheckMate-204 study [4] revolutionized the treatment strategy for this condition. These studies showed that the combination of immunotherapy with ipi/nivo increased clinical response and survival of patients with asymptomatic brain metastases. Consequently, ipi/nivo was recommended as a first-line treatment for patients with asymptomatic melanoma brain metastases in Denmark.

However, patients who are enrolled in trials, such as the aforementioned ABC [3] and CheckMate-204 studies [4], often have a better performance status (PS, according to ECOG) and are generally healthier than real-world patients, which results in trial patients often having a better prognosis and hence superior outcomes compared to real-world patients. 

The aim of this study was to compare clinical benefits, measured in overall survival (OS), Progression-Free Survival (PFS), and complete response (CR), between real-world patients and patients enrolled in clinical trials when administering ipi/nivo in a Danish cohort of previously untreated patients with asymptomatic MBM. 

Such a comparison between real-world data (RWD) and phase II trials allows assessment of applicability to a more diverse population, which is to be encountered in everyday clinical settings. This aids in providing insight into underrepresented patient groups, such as older or frailer individuals, who are often seen by clinicians. This will enhance external validity and potentially contribute to evidence-based clinical practice guidelines.

## 2. Materials and Methods

### 2.1. Participants and Study Design

This retrospective analysis included data derived from the Danish Metastatic Melanoma Database (DAMMED) [5] on Danish patients with melanoma and asymptomatic brain metastases who received ipilimumab/nivolumab from June 2017 to January 2023. The data originated from patients treated at one of the four Danish departments of oncology treating melanoma: Aarhus, Aalborg, Odense, or Herlev.

The data were collected on patients with melanoma and computed tomography (CT) or Magnetic Resonance Imaging (MR-I) verified asymptomatic brain metastases, who were treated with combination immunotherapy (ipi/nivo) as first-line treatment. Data were updated on the 1st of January 2023. Individual patient data were obtained from DAMMED regarding age, gender, melanoma diagnosis, age by first treatment, performance status, treatment institution, Lactate Dehydrogenase (LDH), and therapy outcome measurements. 

A qualitative comparative analysis was then made between the results of the retrospectively collected real-world data from DAMMED and the published results from the ABC [3] and the CheckMate-204 [4] phase II trials. 

### 2.2. Treatments

Danish patients received intravenous ipilimumab 3 mg/kg in combination with nivolumab 1 mg/kg every third week for 4 cycles. Hereafter, nivolumab 3 mg/kg was administered every fourth week. Treatment continued until progression or unacceptable toxicity. Treatment response was assessed every 12 weeks using brain MR-I and positron emission tomography–computed tomography (PET/CT) scans. 

Danish national guidelines recommend a maximum treatment length of two years, with further restrictions for responding patients resulting in shorter treatment durations. If complete response (CR) occurred during treatment, treatment continued for an additional 3 months to ensure confirmation of CR. If partial response (PR) occurred, patients received an additional 6 months of treatment after the best PR was achieved. Treatment was discontinued prematurely if severe toxicity occurred. See Figure 1 below.

### 2.3. Inclusion and Exclusion Criteria

Patients were enrolled in DAMMED and selected using criteria from the Danish Treatment Guidelines for Immunotherapy (1). Furthermore, basic patient inclusion criteria used in the ABC [3] and the CheckMate-204 [4] phase II trials in order to create a similar patient population as seen in Table 1, minimizing bias and confounding. 

Upon enrolment in DAMMED [5], all patients submitted written consent to be included in the database, which can be withdrawn at any time and for any reason. The study was approved by the Regional Listing of Scientific Research in the region of Central Jutland with the file number 1-16-02-50-23.

### 2.4. Participants Identification

After applying the inclusion and exclusion criteria, 79 study participants were found eligible (see Figure 2).

### 2.5. Assessments

Treatment response was determined by radiographic imaging and assessment every 12 weeks while treatment was ongoing. Following Danish national standards, post-treatment patients were scanned every 12 weeks for the first 2 years of follow-up, and thereafter every 6 months for the following 3 years. Brain lesions were assessed using MRI scans to evaluate intracranial response, whereas assessment of extracranial response was done using CT scans. Both were assessed by trained radiologists using the Response Evaluation Criteria in Solid Tumors (RECIST).

Survival was monitored using electronic patient records and the Danish CPR registry during and after the 6-year follow-up period. 

Blood tests were conducted at baseline and prior to each treatment series. Tumor BRAF status was determined by NGS or PCR prior to treatment.

### 2.6. The DAMMED Database

The data were sourced from the DAMMED [5] database, a nationwide Danish database, which prospectively registers treatment for patients suffering from stage III or IV melanoma, covering 95% of Danish patients with melanoma. The dataset was thoroughly checked and corrected for possible errors and inconsistencies.

### 2.7. Outcome Criteria

The primary comparison of interest was the following: PFS, OS, and complete response (CR).

OS was calculated from the first dose of treatment until an event (patient death or censored time point), and PFS was calculated from the first dose to the event (progression or censored time point). OS and PFS were represented graphically using Kaplan–Meier plots to evaluate the outcome. The complete response rate was evaluated by RECIST 1.1 describing the disappearance of all signs of cancer in response to treatment.

### 2.8. Statistical Analysis

Baseline characteristics and study outcomes are stated using descriptive statistics. Survival outcome curves and time-to-event were estimated using the Kaplan–Meier plots with medians presented with a confidence interval (CI) of 95%. Patient demographics included sex, age at MBM diagnosis, melanoma subtype, PS, BRAF mutational status, and LDH level. Statistical analyses were performed using StataSE version 17.

Comparisons between the different cohorts regarding patient characteristics as well as response and survival rates were made using qualitative comparable statistics to analyze the different cohorts’ differences and similarities, as well as the results found by using descriptive statistics.

### 2.9. External Validity 

The results derived from DAMMED and the published data from the ABC [3] and CheckMate-204 [4] phase II trials were compared by examining study design, study population, treatment variation, and outcome measurements to ensure external validity.

## 3. Results

Between 1 June 2017 and 1 January 2023, seventy-nine patients with asymptomatic MBM were treated with ipi/nivo at four different treatment sites in Denmark. 

In the DAMMED cohort, 79 patients received ipi/nivo, with patient characteristics as seen in Table 2. Gender was predominantly male (58%) with a median age of 62 years. The majority (76%) had cutaneous melanoma and a PS 0-1 (96%). BRAF wildtype melanomas were present in 54% of the patients, and 63% had elevated levels of LDH.

Gender distribution differed between the phase II trials and the DAMMED cohort, with the latter having more women (42%) than the ABC [3] (17%) and the CheckMate-204 [4] (33%). 

The ECOG PS was only published for the ABC [3], with 97% of the patients being in PS 0-1. The PS for the Danish cohort was roughly 96% in the 0-1 subgroup and 4% in patients with PS 2. 

The DAMMED cohort had a higher percentage of the BRAF wildtype (54%) than the CheckMate-204 [4] (33%) and ABC (46%) trials. For the ABC [3] study, BRAF V600 mutations were reported in 54% of the patients. The DAMMED cohort had 39%, while the CheckMate-204 [4] (65%) had the highest amount of patients with BRAF V600 mutations. 

When evaluating LDH, 63% had elevated LDH levels in the DAMMED cohort, in contrast to 51% reported in the ABC [3] and 41% in the CheckMate-204 [4].

### 3.1. Response Rates for the DAMMED Cohort

As shown in Table 3, 16.5% of patients achieved CR, while 30.4% achieved PR and 11.4% of patients had SD. A total of 50 (63.3%) patients were still alive after the follow-up period of 5 years, while 29 patients (36.7%) died. It is furthermore notable that 28 patients (35.4%) had to stop treatment due to grade 3 or 4 adverse effects. 

### 3.2. Comparison of Response Rates for the Cohorts

As seen in Table 4, CR was achieved in 16.5% of the patients in the DAMMED cohort, qualitatively compared to 17% for the ABC [3] and CheckMate-204 [4] trials (see Table 4). ORR (47%) resembled the trials, especially the ABC [3] study (46%). The DAMMED cohort had a higher PD (42%) than the CheckMate-204 [4] study (26%), but was similar to the ABC [3] (40%).

When comparing the ABC [3] and the DAMMED cohort, we found a chi^2^ = 0.0123, *p* = 0.912, indicating no statistical difference between the DAMMED cohort and the ABC [3]. Similarly, comparing the CheckMate-204 [4] to the DAMMED cohort yielded a chi^2^ = 0.03834, *p* = 0.536, again signifying a non-significant difference in treatment response between the groups.

### 3.3. Progression-Free and Overall Survival Rates

As presented in Table 5, the PFS rate at 6 months for the DAMMED cohort was 53.5%, which resembled the ABC [3] study (PFS rate 53%). However, the CheckMate-204 [6] found a slightly higher PFS rate of 61.1% at 6 months follow-up. 

In the DAMMED cohort, we found an OS of 83.6% at 6 months. The CheckMate-204 [6] study presented the highest OS rate (92.3%) and had a relatively narrow confidence interval, whereas the ABC [3] study reported an OS rate of 78% with a broader confidence interval (65–94), exhibiting variability in survival outcomes. The median OS rate (mOS) for the DAMMED cohort, the ABC [3] study, and CheckMate-204 [4] was not reached during follow-up.

The CheckMate-204 [4] study from 2021 reported a 36-month overall survival rate of 71.9% and a 6-month PFS rate of 54.1% amongst asymptomatic patients. This resembles the results found in the DAMMED and ABC [3] (cohort A) at 6 months follow-up, where PFS (54%) was equal to that of the CheckMate-204 [4] (54.1%) and ABC [3] study (53%), meaning the CheckMate-204 had a greater PFS at both 6 and 36 months than the ABC and DAMMED at 6 months follow-up.

The median follow-up for the DAMMED cohort was calculated to be 508 days, which is equal to 16.6 months. The median treatment duration was 141 days, which is equal to 4.6 months. The CheckMate-204 [4] trial had an overall median duration of therapy of 3.4 months and a median follow-up of 34.3 months. The ABC [3] study had a median follow-up of 17 months, while the median treatment duration was not reached.

### 3.4. Kaplan Meier Plots for Overall Survival and Progression-Free Survival

The PFS and OS rates for 79 patients with asymptomatic MBM over a 5-year period are presented in Kaplan–Meier plots (Figure 3). The PFS curve rapidly declined during the 1st year after treatment started but stabilized after 18 months. The median PFS was 6.5 months with a CI of 95% (0.39–1.15). A total of 45 patients had progression equal to 58.4% within the follow-up period of 5 years/60 months.

Overall survival declined over the first few years but stabilized after 24 months. Figure 3 shows that mOS had not yet been reached during follow-up. There were 29 (36.7%) reported deaths overall.

## 4. Discussion

The DAMMED cohort was selected using Danish treatment guidelines (1) with similar inclusion and exclusion criteria as those used in the ABC [3] and CheckMate cohorts [4]. This was done to mimic Danish clinical settings with clinical trials. When applying these criteria, we observed similar survival and response rates among patients receiving Ipi/Nivo in both clinical trials and real-world data. However, it is important to acknowledge that because the cohort has been selected, it may not fully reflect real-world conditions. The findings are most directly generalizable to the comparison of OS, PFS, and CR during the studied timeframe. It is important to note that bias cannot be ruled out entirely due to the nonrandomized nature of the DAMMED cohort. Furthermore, these results may not be broadly applicable to other outcome measurements or indications in advanced melanoma or to other databases beyond DAMMED.

Having this in mind, the DAMMED patient data still stemmed from a more diverse population than the clinical trials. This population often includes complex medical histories and comorbidities, and thereby better reflects real-world conditions and demographics. An example of this is the broader age range demonstrated in the DAMMED cohort. Additionally, patients were monitored using electronic patient journals and the CPR registry. This level of detailed monitoring is noteworthy, as it is not available in many countries, allowing for close tracking of treatment outcomes, causes of death, and other critical information.

A comparison between the real-world data (RWD), derived from the DAMMED database, helped ensure that results from clinical trials did not exclude specific patient subgroups. This is because patients excluded from the DAMMED cohort either had symptomatic brain metastasis, or ipi/nivo had not been used as a first-line treatment. This also demonstrates that outcomes found in randomized controlled trials (RCTs) were not solely dependent on the controlled trial setting but could be extrapolated to real-world conditions. Therefore, a comparison between the DAMMED cohort and the two clinical trials [3,4] helped enhance reliability and study effectiveness outside of controlled trial environments.

In summary, this comparison serves to narrow the gap between controlled research settings and clinical complexities, thereby improving the comprehension and application of clinical trial results in everyday clinical settings, giving clinicians ease of mind when choosing a treatment plan for their patients.

The comparison of RWD with phase II trials provides significant clinical value by addressing different factors such as generalizability. Phase II trials normally involve a rather selective and small group of patients, and often have specific inclusion and exclusion criteria, which can limit their applicability to real-world patient cohorts.

The sample sizes differ, with 79 patients in the DAMMED cohort versus 35 patients in the ABC [3] and 101 patients in CheckMate cohort [4]. Such differences in sample sizes can impact the statistical power and reliability of study results, since large sample sizes tend to provide more accurate and representative estimates of population parameters. It may also affect qualitative comparative analysis regarding representativeness, validity, and generalizability.

An important characteristic was performance status (PS), since the better the PS, the better the patients’ general state of health pre-treatment. Patients with a lower PS (0–1) generally have a higher possibility of a positive survival outcome. The PS for the database review was roughly 96% for PS 0–1 and 4% for the PS 2 group. This resembled the ABC [3] study quite well regarding the PS 0–1 with a difference of merely 0.9%.

The median age of the Danish patients was approximately 62 years, a difference of 3 years (5.08%) when compared to the median of 59 years seen in both studies. The DAMMED cohort had the biggest age range (34–82 years), indicating that the age distribution was more spread out and had more variation compared to the studies (ABC 53–68 years, Checkmate 51–66 years). Hence, it provided us with a potentially more generalizable result, since the findings could be applied to a greater age range. Generally, age is a major risk factor for melanoma mortality; therefore, this difference in the age ranges between the groups could be a potential confounder. This is because older patients could have more co-morbidities, and are frailer or less resilient than younger patients. All the aforementioned points could lead to decreased survival rates in RWD.

The DAMMED cohort had a relatively even distribution of men and women, but a higher prevalence of women (41.8%) compared to the ABC [3] (17%) and CheckMate studies [4] (33%). However, data vary on whether female sex is a better prognostic indicator for OS in cancer patients. A study by Joosse et al. [7] found that the potential benefit that female gender might grant became smaller in patients with higher metastatic tumor load. A study by Morgese et al. [8] found that men and women with stage III and IV melanoma did not show a significant sex-based difference when comparing PFS and OS. Therefore, gender was not seen as a confounder of this database review.

When analyzing adverse effects, 35% (28 patients) from the DAMMED database had to stop treatment, due to toxicity. This aligns with a previous meta-analysis [9] which found that approximately 40% of patients receiving ipi/nivo had grade 3 or higher adverse effects.

The biggest advantage of this retrospective study design was its low cost and time efficiency when compared to RCT studies. Generally, RWD such as the data from the DAMMED database [5] are typically also more applicable to a broader population group with the inclusion of a more diverse patient cohort. It features a more pragmatic study design and gives insights into real-world practice patterns, since the data from the DAMMED [5] were collected as part of patients’ routine follow-ups. They thereby reflect real-world clinical conditions in Denmark. Another strength is the possibility of continuing the assessment of the effectiveness and safety of ipi/nivo. These factors all help to enhance the external validity and generalizability of the studies, making RWD such as the data from the DAMMED database [5] an important part of evidence-based medicine [10].

Randomized controlled trials (RCTs) such as the ABC [3] and CheckMate-204 [4] studies are designed to minimize confounding and biases, whereas RWD may be subject to a variety of these.

Overall, trial patients are often healthier and are carefully selected before inclusion to minimize study dropout [11], whereas real-life patients just need to fulfill the requirements set for a specific drug regimen to be administered. Real-life patients therefore often have a greater disease burden and a lower PS. This may result in a general discrepancy between real-world patients and trial patients, due to different inclusion and exclusion criteria.

This paper contains RWD collected from databases where patients were not randomly assigned to treatment groups, and differences in baseline characteristics may vary. Therefore, there might be confounding variables that are not accounted for in the analysis and can affect the validity of causal inferences drawn from the data. This makes it pivotal to ensure that the patient RCTs and the DAMMED are comparable. We excluded two patients with ocular melanoma, since ocular melanoma has different biological behavior to cutaneous melanomas [12], and thereby different treatment strategies. We furthermore mainly excluded patients with symptomatic brain metastasis who had a need for immediate local treatment (n = 114), since these patients would not be applicable for immunotherapy in a real-world clinical setting according to Danish treatment guidelines (1). Patients who had not received ipi/nivo as a first-line treatment (n = 33) were also excluded to avoid confounding.

A 2022 retrospective study by D. Kuzmanovszki et al. [13] evaluated the efficacy of treatment on real-world patients with advanced melanoma without brain metastasis treated with Nivolumab or Pembrolizumab monotherapy. Data were compared with the Checkmate-066 phase III trial [14]. The data suggested the survival benefit from monotherapy in real-world patients with melanoma was similar to the findings in the phase III trial.

The phase III CheckMate-067 trial [2] evaluated the efficacy of treatment in patients with metastatic melanoma without brain metastasis administered with a combination of ipi/nivo vs. nivolumab and vs. ipilimumab. The study found a mOS of 72.1 months and a median PFS (mPFS) of 11.5 months. At 36 months, PFS was 39% and OS was 68%. The CheckMate-204 [4] study reported a 36-month PFS rate of 54.1% and an OS rate of 71.9% among asymptomatic patients with brain metastases.

Upon comparison of the Checkmate-067 trial with the data from DAMMED, the OS at 6 months was approximately 84%, whereas the CheckMate-67 reported a 6-month OS of approximately 84.4%. The DAMMED data showed a 6-month PFS of 53.5%, while Checkmate-67 reported a 6-month PFS of 55.7%. The mPFS in the CheckMate-67 was 11.5 months, whereas this was not yet reached in the DAMMED data analysis. This demonstrates similar results regarding survival for patients with melanoma with brain metastasis.

## 5. Conclusions

When performing the qualitative comparison of the DAMMED cohort with the two clinical trials, we observed relatively similar baseline patient characteristics as well as relatively similar response and survival rates. The findings underscored the reproducibility of clinical trial results in real-world conditions.

In conclusion, this RWD study showed similar efficacy of combination immunotherapy for asymptomatic MBM compared to previously published RCTs. This holds true for both patients eligible or ineligible for participation in the ABC and CheckMate-204 studies. Hence, the results from the RCTs are comparable to RWD, showing no significant difference in outcome measured in OS, PFS, and CR upon comparison.

## Figures and Tables

**Figure 1 cancers-16-02559-f001:**
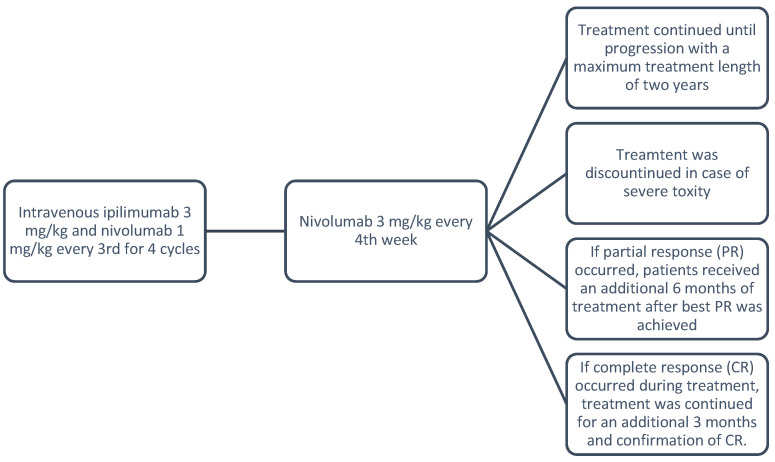
Treatment protocol.

**Figure 2 cancers-16-02559-f002:**
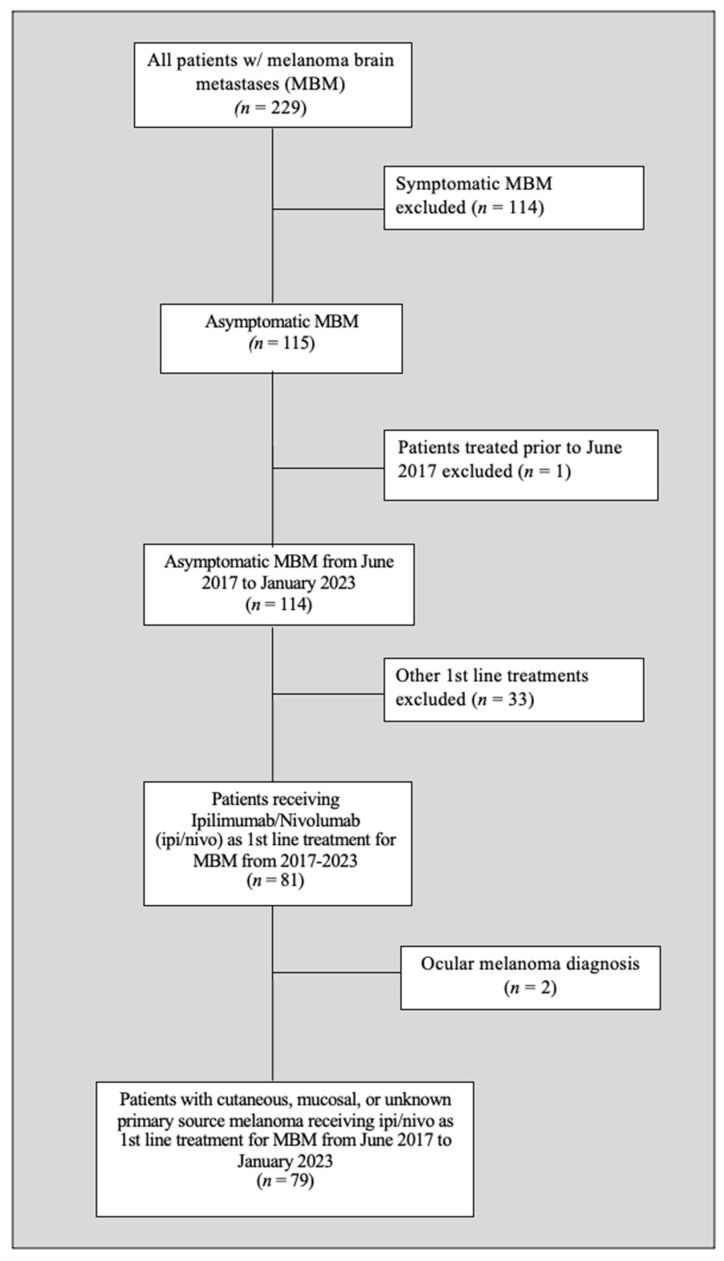
Patient selection flow chart.

**Figure 3 cancers-16-02559-f003:**
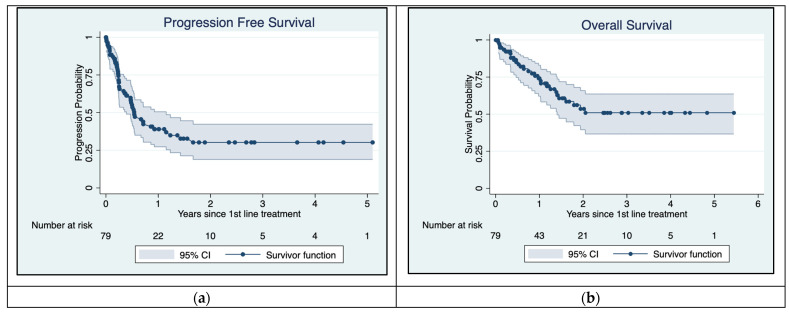
Kaplan–Meier plots for Progression-Free Survival (**a**) and overall survival (**b**). Median PFS is approximately 6.5 months. Median overall survival has not yet been reached.

**Table 1 cancers-16-02559-t001:** Inclusion and exclusion criteria.

Inclusion Criteria
Age ≥ 18 years *
ECOG PS 0–2
1st line treatment, ipilimumab/nivolumab, for MBM **
≥1 measurable brain metastasis assessed by MRI or CT
Exclusion criteria
Previous BRAF- and/or MEK-inhibitor treatment or other antineoplastic treatments. Treatment with corticosteroid or other immunosuppressive agents within the last 14 days before initiation of immunotherapy.
Previous local brain therapy (stereotactic radiosurgery or surgery) within three weeks before initiation of immunotherapy.
Symptomatic brain metastases.
Requiring immediate local treatment.
Patients who had previously received organ transplants.
Ocular melanoma.

* No upper age limit if PS was met. ** Previous adjuvant nivolumab treatment accepted if ipi/nivo is 1. Line treatment for metastatic melanoma.

**Table 2 cancers-16-02559-t002:** Baseline patient characteristics for patients from the DAMMED, ABC study, and CheckMate-204 study.

Patient Characteristics	DAMMED	ABC [3] (Cohort A)	CheckMate-204 [4] (Asymptomatic Cohort)
Sample size	*n* = 79	*n* = 35	*n* = 101
Sex			
- Male	46 (58.2)	29 (83)	68 (67)
- Female	33 (41.8)	6 (17)	33 (33)
Mean age at first treatment (range)—years	61.7 (34–82)	59 (53–68)	59 (51–66)
Performance Status (ECOG)			Data not published
0–1	76 (96.2)	PS 0–1: 34 (97)	
2	3 (3.8)	PS 2: 1 (3)	
BRAF mutation status			
- Wild type	43 (54.4)	Wild type: 16 (46)	Wild type: 33 (33)
- V600 mutations *	31 (39.2)	V600 mutation: 19 (54)	Mutant: 66 (65)
- Other	5 (6.4)		Na: 2 (2)
LDH:			
- >ULN	50 (63.3)	LDH increase:	>ULN: 41 (41)
- <ULN	27 (34.2)	18 (51)	<ULN: 60 (59)
- ND	2 (2.5)		

* V600 mutations included V600E, V600X and V600K; ND: non-determined; Na: Not reported; ULN: Upper Limit of Normal.

**Table 3 cancers-16-02559-t003:** Total response rates from DAMMED.

Response Rates	*n* (%) *n* = 79
Total response	
- CR	13 (16.5)
- PR	24 (30.4)
- SD	9 (11.4)
- PD	33 (41.8)
Deaths in total	29 (36.7)
Alive	50 (63.3)
Reasons for treatment discontinuation	
- Progression	24 (30.4)
- Toxicity, investigator’s choice	28 (35.4)
- Other, investigator’s choice	2 (2.5)

**Table 4 cancers-16-02559-t004:** Response rates.

Response Rates *n* (%)	DAMMED *n* = 79	ABC [3] (Cohort A) *n* = 35	Checkmate-204 [4] *n* = 101
Response			
- CR	13 (16.5)	6 (17)	17 (17)
- PR	24 (30.4)	10 (29)	35 (35)
- SD	9 (11.4)	4 (11)	4 (4)
- PD	33 (41.8)	14 (40)	26 (26)
- Na *	0 (0)	1 (3)	19 (19)
- ORR	37 (46.9)	16 (46)	52 (52)
- *p*-value **	*	0.912	0.536

* Non-evaluable; ** *p*-value for the Pearson’s chi-square test.

**Table 5 cancers-16-02559-t005:** Survival statistics at 6 months follow-up.

Survival Statistics *n* (%)	DAMMED *n* = 79	ABC [3] (Cohort A) *n* = 35	CheckMate-204 Study, 2018 [6] *n* = 101
- 6-month PFS rate	53.5% (41–64) *	53% I 51% **	61.1% (50.0–70.5) *
- 6-month OS rate	83.6% (73–90) *	78% (65–94)	92.3% (84.5–96.3) *
- Median Treatment Duration (months) ***	4.6 months	NR (8.5-NR)	3.4 months

* Global response; ** PFS for intracranial I extracranial disease; *** Median time between first dose date and the date of death or last known date alive.

## Data Availability

Due to legal restrictions under Danish legislation concerning data privacy and protection, the data supporting the findings of the study cannot be shared publicly. Interested researchers may contact the corresponding authors for further information or reach out to the Regional Listing of Scientific Research regarding file number 1-16-02-50-23.

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
