# Peer review of "Efficacy of Ipilimumab and Nivolumab in Patients with Melanoma and Brain Metastases—A Danish Real-World Cohort"

_cancers, 2024, doi:10.3390/cancers16142559_

Round 1

Reviewer 1 Report

Comments and Suggestions for Authors

The authors  aimed to compare the clinical benefits between real-world patients and patients enrolled in clinical trials when administering ipilimumab/nivolumab in treatment patients with asymptomatic melanoma and asymptomatic brain metastases.  According to the study results, combined immunotherapy has been recommended as first-line treatment for patients with asymptomatic melanoma brain metastases.

- Line 74 and line 155. Which info is correct?

-I suggest that the examination design (2.2. Treatments) should be presented as a table. 

- Please correct the line 339: by D. Kuzmanovszki 

- Conclusion section should not contain citiations. Please remove them.

Comments on the Quality of English Language

- Line 28-29, 37: naïve? Please correct as naive.

-Lines 102-103 'Danish treatment guideline for immunotherapy' correct as Danish Treatment Guideline for Immunotherapy.

Author Response

Dear reviewers

I’m writing to express my gratitude for taking the time to review the article “Efficacy of ipilimumab and nivolumab in patients with melanoma and brain metastases - a Danish real-world cohort”.  Your expertise and insight are very valuable to us, and I greatly appreciate your valuable feedback. We have carefully considered all your comments and suggestions and have improved the quality and clarity of the manuscript.

In response to the reviewers’ comments, we have made the following changes:

1. Enhanced the methodology section, now including a section about the DAMMED database, a figure regarding the treatment protocol and clarification regarding survival monitoring.

2. Expanded the results section with new statistical analyses to strengthen our findings.

3. Improved the discussion regarding strengths and limitations.

Comments for reviewer #1:

1. Line 74 and line 155. Which info is correct?

• This has now been corrected, see line 74 and 167

2. I suggest that the examination design (2.2. Treatments) should be presented as a table.

• This has been corrected and can be seen as figure 1.

3. Please correct the line 339: by D. Kuzmanovszki:

• Corrected, see line 360.

4. Conclusion section should not contain citiations. Please remove them.:

• Corrected, see section 5. conclusion.

5. Line 28-29, 37: naïve? Please correct as naive.:

• Corrected, see line 28 and 37

6. Lines 102-103 'Danish treatment guideline for immunotherapy' correct as Danish Treatment Guideline for Immunotherapy.:

• Corrected, see line 107

Comments for reviewer #2:

1) Thank you for this good suggestion to help strengthen our findings. A chi-squared test has been performed and added under section 3.3. table 4 and line 211-215.

However, it was not possible to include a log-rank test, since we did not have access to some of the raw data from the ABC- and Checkmate-204 study.

2) New paragraph has been added to section 2.6, line 136-140 about the DAMMED database. Furthermore, there has been made additions to paragraph 2.5 to included follow up and survival monitoring (line 131-132). Measurements to avoid confounding can be read in the discussion section.

Thank you once again for your consideration and time in reviewing our manuscript. We look forward to your positive feedback.

Kind regards,
Karoline Dreyer Kattenhøj and Henrik Schmidt.
Faculty of Health, Aarhus University and the Department of Oncology, Aarhus university hospital

Reviewer 2 Report

Comments and Suggestions for Authors

Hi, Karoline D. Kattenhøj, 

Thank you very much for submitting your manuscript to our journal, your paper is very creative and interesting. Your method is comparing the data from clinic trail with results from Danish Metastatic Melanoma Database, which shown that Complete Response (CR), Progression Free survival (PFS) and Overall Survival (OS) rates of asymptomatic melanoma brain metastases (MBM) treated with first line ipilimumab and nivolumab demonstrates are comparable to previously published phase II trials.  

This paper suggests that combination immunotherapy can be recommended as first line treatment for patients with asymptomatic, treatment naïve melanoma brain metastases. I believe your paper is wonderful and reliable. I would suggest the journal accept your manuscript.

Author Response

Dear reviewers

I’m writing to express my gratitude for taking the time to review the article “Efficacy of ipilimumab and nivolumab in patients with melanoma and brain metastases - a Danish real-world cohort”.  Your expertise and insight are very valuable to us, and I greatly appreciate your valuable feedback. We have carefully considered all your comments and suggestions and have improved the quality and clarity of the manuscript.

In response to the reviewers’ comments, we have made the following changes:

1. Enhanced the methodology section, now including a section about the DAMMED database, a figure regarding the treatment protocol and clarification regarding survival monitoring.

2. Expanded the results section with new statistical analyses to strengthen our findings.

3. Improved the discussion regarding strengths and limitations.

Comments for reviewer #2:

1) Thank you for this good suggestion to help strengthen our findings. A chi-squared test has been performed and added under section 3.3. table 4 and line 211-215.

However, it was not possible to include a log-rank test, since we did not have access to some of the raw data from the ABC- and Checkmate-204 study.

2) New paragraph has been added to section 2.6, line 136-140 about the DAMMED database. Furthermore, there has been made additions to paragraph 2.5 to included follow up and survival monitoring (line 131-132). Measurements to avoid confounding can be read in the discussion section.

Thank you once again for your consideration and time in reviewing our manuscript. We look forward to your positive feedback.

Kind regards,

Karoline Dreyer Kattenhøj and Henrik Schmidt.
Faculty of Health, Aarhus University and the Department of Oncology, Aarhus university hospital

Reviewer 3 Report

Comments and Suggestions for Authors

Dear Authors,

I have reviewed your manuscript titled "Efficacy of ipilimumab and nivolumab in patients with melanoma and brain metastases: a Danish real-world cohort" with great interest. Your study addresses a critical and timely topic in oncology, particularly the application of combination immunotherapy in a real-world setting for patients with melanoma and brain metastases. Below, I provide my detailed comments and suggestions for improving the manuscript:

1- The methods section can be improved by including a detailed description of the statistical methods used for data analysis, particularly any adjustments made for potential confounding variables. Clarifying the follow-up procedures and how consistent follow-up was ensured across different treatment centers would also add to the reliability of the findings.

2- The presentation of the results can be improved by providing additional comparision with previously published data. For instance, the study's reported 16.5% Complete Response (CR) rate and 46.9% Overall Response Rate (ORR) should be compared with the CR and ORR rates in the ABC (17% CR, 46% ORR) and CheckMate-204 (17% CR, 52% ORR) trials. Given the close percentages, it's likely that the differences in CR rates between the study and the ABC and CheckMate-204 trials are not statistically significant. The ORR difference between the study and CheckMate-204 might be statistically significant but would require a chi-squared test to confirm.

Similarly, the 6-month Progression-Free Survival (PFS) rate of 53.5% and Overall Survival (OS) rate of 83.6% should be contextualized against the 53% PFS and 78% OS rates from the ABC study, and the 61.1% PFS and 92.3% OS rates from CheckMate-204. The PFS rates between the study and the ABC trial are very close, suggesting no significant difference. However, the PFS rate difference between the study and CheckMate-204 may be statistically significant. Similarly, the OS rate differences would require a log-rank test to determine significance.

Author Response

(The authors gave the same response as above.)
